# Tolerance of Prolonged Oral Tedizolid for Prosthetic Joint Infections: Results of a Multicentre Prospective Study

**DOI:** 10.3390/antibiotics10010004

**Published:** 2020-12-23

**Authors:** Eric Senneville, Aurélien Dinh, Tristan Ferry, Eric Beltrand, Nicolas Blondiaux, Olivier Robineau

**Affiliations:** 1Infectious Diseases Department, Gustave Dron Hospital, 59200 Tourcoing, France; orobineau@ch-tourcoing.fr; 2Faculty of Medicine Henri Warembourg, Lille University, 59000 Lille, France; 3French National Referent Centre for Complex Bone and Joint Infections, CRIOAC Lille-Tourcoing, 59000 Lille, France; ebeltrand@ch-tourcoing.fr (E.B.); nblondiaux@ch-tourcoing.fr (N.B.); 4Infectious Diseases Department, Ambroise Paré Hospital, 92100 Boulogne-Billancourt, France; aurelien.dinh@aphp.fr; 5French National Referent Centre for Complex Bone and Joint Infections, CRIOAC Paris-Ambroise, 75000 Paré, France; 6Infectious Diseases Department, Croix-Rousse Hospital, 69004 Lyon, France; tristan.ferry@univ-lyon1.fr; 7French National Referent Centre for Complex Bone and Joint Infections, CRIOAC Lyon, 69004 Lyon, France; 8Orthopaedic Surgery Department, G. Dron Hospital Tourcoing, 59200 Tourcoing, France; 9Microbiology Laboratory, G. Dron Hospital Tourcoing, 59200 Tourcoing, France

**Keywords:** tedizolid, prosthetic joint infections, prolonged oral treatment, tolerance, compliance

## Abstract

Objectives: Data on clinical and biological tolerance of tedizolid (TZD) prolonged therapy are lacking. Methods: We conducted a prospective multicentre study including patients with prosthetic joint infections (PJIs) who were treated for at least 6 weeks but not more than 12 weeks. Results: Thirty-three adult patients of mean age 73.3 ± 10.5 years, with PJI including hip (*n* = 19), knee (*n* = 13) and shoulder (*n* = 1) were included. All patients were operated, with retention of the infected implants and one/two stage-replacements in 11 (33.3%) and 17/5 (51.5%/15.2%), respectively. Staphylococci and enterococci were the most prevalent bacteria identified. The mean duration of TZD therapy was 8.0 ± 3.27 weeks (6–12). TZD was associated with another antibiotic in 18 patients (54.5%), including rifampicin in 16 cases (48.5). Six patients (18.2%) had to stop TZD therapy prematurely because of intolerance which was potentially attributable to TZD (*n* = 2), early failure of PJI treatment (*n* = 2) or severe anaemia due to bleeding (*n* = 2). Regarding compliance with TZD therapy, no cases of two or more omissions of medication intake were recorded during the whole TZD treatment duration. Conclusions: These results suggest good compliance and a favourable safety profile of TZD, providing evidence of the potential benefit of the use of this agent for the antibiotic treatment of PJIs.

## 1. Introduction

Prosthetic joint infection (PJI) is a serious and complex complication following arthroplasty at an incidence rate after hip or knee replacement of 1 to 2% [1]. The aims of the management of patients with PJIs are to restore satisfactory joint function and to eliminate infection. Surgical options include debridement antibiotics and implant retention (DAIR), one- or two-stage replacement, arthroplastic resection and, sometimes, amputation. Given the increasing burden of these infections, especially among the elderly population, developing new therapies such as cell therapy to prevent the progression of osteo-arthritis, and thus, the need for total joint arthroplasty, is an important field of research [2,3,4,5]. The antibiotic treatment of patients with PJIs is limited by the tolerance of its prolonged administration and the resistance level of some pathogens [6,7]. Gram-positive cocci, especially coagulase-negative staphylococci (CoNS), are predominant bacteria which are responsible for infections in and around orthopaedic devices [8]. In this context, the use of the oxazolidinone agent linezolid (LZD) has been validated, but potential bone-marrow, neurologic, and metabolic toxicity limit treatment duration to no more than two to three weeks [9,10,11]. Additionally, the wide use of LZD has resulted in the emergence of CoNS carrying *cfr* genes which are responsible for high levels of LZD resistance [12]. The combination of rifampicin with LZD leads to a reduction in LZD blood concentration, which is associated with a lower rate of adverse hematologic effects but also with lower clinical remission rates [13,14]. Tedizolid (TZD) phosphate is a second generation oxazolidinone which is indicated for the treatment of acute bacterial skin and skin structure infections in adults [15,16,17,18]. In the Establish 1 and 2 studies, gastrointestinal disorders and bone marrow toxicity were less frequent in TZD than in LZD patients [19,20]. However, the duration of TZD treatment did not exceed six days. TZD has a high oral bioavailability and can be administered once-daily; furthermore, drug–drug interactions with mono-amine oxidase inhibitors (MAOI), serotonin-reuptake inhibitors (SRI) or rifampicin are unlikely, although the latter has recently been questioned [21,22]. Recent in vitro and animal studies suggested that the addition of rifampicin to TZD was likely to achieve a synergistic effect against methicillin-resistant *Staphylococcus aureus* (MRSA) and *S. epidermidis*, and prevent the emergence of rifampicin-resistant mutants [23,24]. While TZD appears to be an attractive candidate for the treatment of PJIs due to gram-positive cocci, and has shown satisfactory efficacy and tolerability in clinical trials, data about its tolerability and compliance in long-term treatments are lacking. The aim of the present multicentre prospective cohort study was to assess the long-term safety profile and compliance of oral TZD in monotherapy or in combination therapy for the treatment of PJIs.

## 2. Results

Thirty-three adult patients (sex ratio female/male 17/16) of mean age 73.3 ± 10.5 years were included from August 2018 to November 2019. A total of 17 patients (51.5%) were enrolled at the Tourcoing Centre, 13 (39.4%) at the Ambroise Paré Centre and 3 (9.1%) at the Lyon Centre. Patient characteristics are presented in Table 1. Despite chronic infection, three patients were treated with a debridement antibiotic and implant retention (DAIR) because of their age and general status which contraindicated the replacement of the implant. TZD use was used to avoid LZD potential toxicities or drug–drug interactions in 16 patients (48.5%), or because of previous, LZD-related adverse events in three patients (9.1%). Among these three patients, one had experienced thrombocytopenia, one anaemia and one gastro-intestinal intolerance. No included patients were receiving MAOI or SRI concomitantly with TZD therapy. Staphylococci were the most prevalent bacterium identified in our patients, accounting for 58% of the total number, and including 21 (42%) methicillin-resistant strains (Table 2). Infection was polymicrobial in 18 cases (54.5%) among which five were associated with gram-negative rods, all of which were susceptible to fluoroquinolones. Geometric mean MIC values for linezolid, as determined by E-test methods, were 1.24 ± 0.83 mg/L, 1.5 ± 0.7 mg/L and 1.64 ± 0.48 mg/L for *Staphylococcus* spp., *Streptococcus* spp. and *Enterococcus* spp., respectively. MIC measurements or TZD blood levels were not routinely performed for tedizolid in this study.

Following postoperative empirical antibiotic therapy (PEAT) of median duration of 7 days (range 4 to 14 days), the mean duration of TZD therapy was 8.0 ± 3.27 weeks (ranging from 6–12 weeks). Among the 27 out of 33 patients (81.8%) who completed the planned therapy, the mean duration of TZD was 8.77 weeks ± 2.79 (range 6–12 weeks). The mean total duration of the antibiotic treatment including PEAT and targeted therapy was 9.15 ± 3.43 weeks (7–12). TZD was associated with another antibiotic in 18 patients (54.5%), e.g., rifampicin in 16 cases (48.5%).

In total, 20 patients (60.6%) experienced at least one adverse event during TZD therapy. A list of adverse events (AE) potentially attributable to tedizolid is shown in Table 3; the most frequent AE were anaemia (*n* = 4) and pruritus (*n* = 4). Six patients (18.2%) had to stop TZD therapy prematurely because of (i) intolerance which was potentially attributable to TZD (*n* = 2), (ii) early failure of PJI treatment (*n* = 2) or (iii) severe anaemia (*n* = 2). TZD-attributable discontinuation episodes consisted of inflammatory arthritis of the wrist and knee in one patient who also received doxycycline and did not improve after stopping doxycycline but partially recovered after discontinuation of TZD, and vomiting in another patient who received TZD alone (Appendix A
Table A1). According to the definitions used to describe the bone marrow toxicity profile of TZD, 8 patients (24.2%) experienced haematological adverse events including anaemia in 4 cases, 2 of which presented acute haemorrhage, leukopenia in 2 cases and thrombocytopenia in 2 cases (Table A1). With the exception of the two patients with acute haemorrhage, none of these adverse events resulted in withdrawing TZD therapy. Although a gastric haemorrhage in one patient and a hematoma at the surgical site in another patient resulted in acute severe anaemia which was most probably unrelated to TZD therapy, the treatment was discontinued. Haematological adverse events were mild and resolved spontaneously during TZD therapy except in the two patients with severe anaemia who received a blood transfusion. Non haematological adverse events were recorded in 13 (39.4%) patients for whom no premature discontinuation of TZD therapy was required, and were mostly pruritus (*n* = 4), headache (*n* = 2) and insomnia (*n* = 2) (Table A1). There was no safety signal for TZD-associated optic or peripheral neurologic toxicity or metabolic disorder. Overall, the proportion of patients who experienced TZD-attributable adverse event did not differ significantly in patients treated with a combination of antibiotics or with TZD alone [13/18 (72.2%) versus 8/15 (52.3%), respectively; *p* = 0.45], nor did it vary according to the use of rifampicin in combination with TZD or the total duration of TZD therapy (Table 4).

The follow-up of the haematological parameters showed a significant increase of haemoglobin blood levels between baseline and week 6 followed by stabilisation, as well as a significant decrease in platelets, leukocytes and neutrophils counts between baseline and week 6 followed by stabilisation until the end of the treatment (Figure 1A–D).

Regarding compliance to TZD therapy, no cases of two or more omissions of medication intake during TZD treatment were recorded, in accordance with the number of pills present in the returned boxes. Six failures (18.2%), including two early cases, were recorded at one-year following the end of TZD therapy.

## 3. Discussion

We report the first prospective cohort study to date providing data on the safety and compliance of prolonged use (i.e., ≥6 weeks) of oral TZD phosphate at a 200-mg, once-daily dose for the treatment of PJIs. As our safety results suggest, oral TZD therapy administered for 6 to 12 weeks according to the current recommendations [25] can be considered for the treatment of PJIs. The overall proportion of patients who experienced an adverse event (60.6%) may appear high, but this may be explained by the design of the study which allowed us to report an exhaustive list of adverse events. Despite the nonoptimal profile regarding the general status of our patients, the tolerance of prolonged oral TZD therapy allowed us to complete the therapy in more than 80% of the cases. Indeed, 19 patients (57.6%) had comorbidities and 27 (81.2%) had an ASA score ≥ 2, which is significantly different from the populations of patients evaluated in other, pivotal clinical trials [19,20]. Our results are close to those reported by Kim et al. on a series of 25 patients with nontuberculous mycobacterial infections treated with a median duration of TZD therapy of 91 days [26]. Eleven of their patients (44%) experienced an adverse event including gastrointestinal intolerance in five patients (20%) and thrombocytopenia in one (4%); no case of anaemia was recorded, while peripheral neuropathy was reported in five patients (20%). The attribution of an adverse event to TZD was, however, difficult, as almost all patients were receiving multidrug therapy. The mean age of our patients was 73.3 years, which is quite high with regard to the risk of developing bone marrow toxicity to LZD, as reported by several authors [9,10]. The correction of LZD-induced bone marrow toxicity after switching to TZD observed in one of our patients has already been reported elsewhere [27,28]. Overall, we only recorded a few significant haematological abnormalities which did not result in discontinuation of TZD therapy, except in patients with acute haemorrhage. There were no differences in safety, especially with regard to haematological laboratory changes, between patients receiving TZD in combination with rifampicin versus patients receiving TZD alone (full data are available upon request), as reported with LZD-rifampicin combination [9,10]. We hypothesise that the increase of haemoglobin values during TZD treatment represents a restoration process after blood spoliation secondary to the surgical intervention, while the decrease of platelets and WBC during TZD treatment might be related to the resolution of the infectious process. The incidence of digestive disorders reported in our patients is close to the results of a meta-analysis by Lan et al., noting, however, that the duration of TZD therapy in the studies included was six days, as recommended for the treatment of acute bacterial skin and skin structure infections [29].

The main limitations of the present pilot study are the small size of the studied population and the assessment of the patients’ adherence to TDZ treatment, which was based on the return of the pillboxes and on patient self-reporting. The strengths of the present study are its prospective design and the selection criteria which allowed investigators to include patients in a real-life setting. We strongly believe that the inclusion of patients, regardless of age and risk factors for bone marrow toxicity, enhanced the external validity of our conclusions regarding the tolerance of prolonged oral TZD therapy in patients treated with PJIs.

## 4. Materials and Methods

The purpose of the present study was to obtain reliable data on the tolerance, compliance and efficacy of prolonged (i.e., ≥6 weeks) use of TZD alone or in combination therapy for the treatment of PJIs. We present herein data about adherence and tolerance. As post-treatment follow-ups are currently underway, we present data only for the one-year follow-up. We conducted a prospective multicentre cohort study in three French national centres for the management of complex bone and joint infections (also called CRIOAc): Lille-Tourcoing, Paris-Ambroise Paré and Lyon [30].

### 4.1. Definitions

Adult patients with PJIs defined according to the MSIS 2018 [31] criteria and for whom TZD treatment was indicated according to the investigator’s decision were prospectively included. All patients gave their written informed consent after an explanation of the protocol by the investigating physician. PJIs were characterised according to: acute haematogenous (infection with three-week duration or less of symptoms after an uneventful postoperative period), early postinterventional (infection that manifested within one month after implantation) and chronic (infection with symptoms that persisted for >3 weeks, i.e., beyond the early postinterventional period) according to Zimmerli’s definition [32].

Patients demographic data (age, gender, body mass index), comorbidities, microbiology, prior use of LZD and reason for TZD use (e.g., failure and/or toxicity of previous treatment, need to avoid linezolid toxicity or drug–drug interactions), treatment duration, concomitant antibiotics, potential adverse events attributable to TZD were recorded. Laboratory data were recorded at baseline, weekly and at the end of treatment, including haemoglobin, white blood count (WBC), platelet count, alanine aminotransferase (ALT), aspartate aminotransferase (AST) and Protein-C reactive (PCR).

Clinically significant laboratory changes were defined as: (1) anaemia, decrease in haemoglobin ≥2 g/L from baseline after TZD initiation and classified as severe if haemoglobin was <8 g/dL, (2) leukopenia, white blood count (WBC) of <4 G/L after TZD initiation and classified as mild (low limit normal to 3 G/L), moderate (≥2-<3 G/L) and severe (<2 G/L), (3), neutropenia, absolute neutrophil count of <1.5 G/L after TZD initiation, (4) thrombocytopenia, platelet count of <150 G/L after TZD initiation and classified as mild (75–150 G/L), moderate (50-<75 G/L) or severe (<50 G/L); for patients with a baseline platelet count of <150 G/L, thrombocytopenia was defined as a reduction of 25% from the baseline, and (5) elevated AST or ALT 3 times above the upper limit of normal.

Microbiological documentation was based on joint aspiration and/or intraoperative culture samples. During surgical procedures, at least three tissue samples were taken in different areas suspected of infection, using a separate sterile instrument for each sample. The antibiotic susceptibility profile of all pathogens was assessed either by the Vitek 2 cards (BioMérieux, Marcy l’Etoile, France) or by agar diffusion technique using the procedure and interpretation criteria proposed by the Comité de l’Antibiogramme de la Société Française de Microbiologie (CA-SFM EUCAST 2018) (http://www.sfm-microbiologie.org). Methicillin resistance was confirmed by the detection of the *mecA* gene if required.

Adverse events (AEs) were identified from patients’ medical records and laboratory data. Associations of adverse events with TZD and related antibiotics were assessed as suggested by the patient’s physician and confirmed by the principal investigator according to the chronology of events, as were the need to reduce the daily dosage of the potentially problematic antibiotic, data from any attempt to reintroduce such a mode of treatment, and the type of recorded toxicity (e.g., anaemia, thrombocytopenia and peripheral neuropathy for TZD, tendonitis, and myalgia for levofloxacin and drug–drug interaction for rifampicin). To be attributable to a given antibiotic, a reduction in the daily dosage and/or discontinuation due to intolerance had to be recorded as well as temporal association with event resolution after discontinuation or dose reduction of the agent in question.

### 4.2. Antibiotic Treatment

TZD was administered orally at a once daily dose of 200 mg (i.e., one tablet) as a single antibiotic therapy or in combination therapy with another agent with proven activity against the involved pathogen(s) according to the physician’s choice. The duration of antibiotic therapy ranged from 6 to 12 weeks. Exclusion criteria were pregnant women or women of childbearing age who were not using contraception, breastfeeding intolerance to TZD, allergy to oxazolidinone, the detection of bacteria which were nonsusceptible to TZD, patients with uncertainty regarding the possibility of achieving a one-year follow-up after the end of treatment or the absence of written consent. Patients were examined during consultations every 3 weeks during treatment and at 6 months and one year after the EOT. During treatment, special attention was paid to potential neurological and optical side effects, as well as to possible drug–drug interactions.

### 4.3. Statistics

Data are presented as numbers (percentages) for qualitative variables and as medians (interquartile range: IQR) or means (SD) for quantitative variables. We compared biological variable that might have been affected by the use of oxazolidinone between baseline and day 42 and between day 49 and day 84 using the Student *t*-test, with *p* = 0.05 being set as the threshold of significance.

### 4.4. Ethics

Research was conducted in accordance with the Declaration of Helsinki and national and institutional standards. The study was recorded on clinicaltrial.gov under the number NCT03378427, in the EudraCT database under the number 2017-001238-24 and was approved by the French Sud Mediterranean IV *Committee* of *Protection* of the *People* in Biomedical Research on 21 November 2017 under the number 17 10 09. This interventional survey was declared to the National Agency for Medicines and Safety of Health Products under the number 17060A-43. Tedizolid was supplied by *Merck Sharp & Dohme*, Inc.

## 5. Conclusions

The results of the present study suggest good compliance and a favourable safety profile of TZD, providing evidence of the potential benefit of the use of this agent for the antibiotic treatment of PJIs.

## Figures and Tables

**Figure 1 antibiotics-10-00004-f001:**
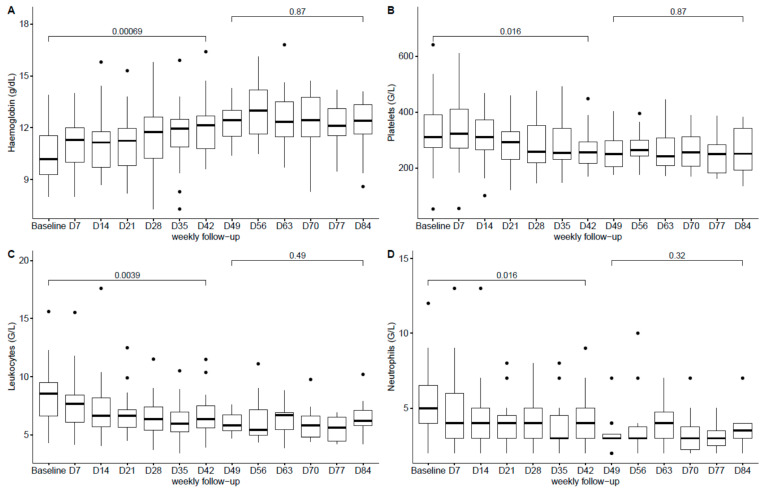
Boxplot of the haematological parameters ((**A**): Haemoglobin, (**B**): Platelets, (**C**): Leukocytes, (**D**): Neutrophils) during Tedizolid therapy. Each box represents median, interquartile range, largest, smallest and outside (point) values.

**Table 1 antibiotics-10-00004-t001:** Demographic data of 33 patients with periprosthetic joint infections.

Patient Characteristics	Values and Number of Patients (%)
Age, years in mean ± SD	73.3 ± 10.5
Sex ratio (female/male)	17/16
Body mass index, kg/m^2^ mean ± SD (>30)	29.7 ± 6.2 (51.5)
Comorbidities * - Diabetes mellitus - Cancer - Liver cirrhosis - Chronic obstructive pulmonary disease - Rheumatoid polyarthritis - Chronic renal failure	19 (57.6) 10 (30.3) 5 (15.2) 1 (3.0) 4 (12.1) 1 (3.0) 2 (6.0)
American Society of Anaesthesiologists score ≥2 [range]	27 (81.2) [1–3]
Previous surgical revision of the prosthesis ≥1 [range]	12 (36.4) [1–10]
Total joint arthroplasty - Total hip prosthesis - Total knee prosthesis - Total shoulder prosthesis	19 (57.6) 13 (39.4) 1 (3.0)
Age of the prosthesis, months mean ± SD [range]	24.5 ± 39.0 [1–180]
Type of infection - Early postinterventional - Chronic - Acute haematogenous	6 (18.2) 25 (75.8) 2 (6.1)
Surgical intervention - Drainage and retention of the implant - One-stage replacement - Two-stage replacement	11 (33.3) 17 (51.5) 5 (15.2)
Fever (temperature > 38.0 °C)	4 (12.1)
Fistula	12 (36.4)
C-reactive protein at baseline, mg/L mean ± SD [range; IQR]	42.16 ± 34.9 [5.8–111; 52]
White blood cells at baseline, G/L mean ± SD [range; IQR]	8.34 ± 2.5 [4.3–15.6; 3.2]

SD: standard deviation. *: 4 patients had ≥2 comorbidities.

**Table 2 antibiotics-10-00004-t002:** Microbiology of 33 patients with periprosthetic joint infections.

Bacteria	N° of Strains (%)
Gram positive cocci	43 (86.0)
- *Staphylococcus aureus* (MRSA = 7)	13 (26)
- *Staphylococcus epidermidis* (MRSE = 14)	15 (30)
- *Staphylococcus caprae* (MR = 2)	1 (2)
- *Streptococcus agalactiae*	2 (4)
- *Corynebacterium striatum*	4 (8)
- *Enterococcus faecalis*	7 (14)
- *Enterococcus gallinarum*	1 (2)
Gram negative bacilli	5 (10)
- *Escherichia coli*	2 (4)
- *Klebsiella pneumoniae*	1 (2)
- *Pseudomonas aeruginosa*	1 (2)
- *Pasteurella multocida*	1 (2)
Anaerobes	2 (4)
- *Cutibacterium acnes*	2 (4)
Total number of bacterial strains	50 (100)

Legend: MRSA: Methicillin-resistant *Staphylococcus aureus*; MRSE: Methicillin-resistant *Staphylococcus epidermidis*; MR: Methicillin-resistant.

**Table 3 antibiotics-10-00004-t003:** Episodes of adverse effects reported in 33 patients during tedizolid therapy.

Adverse Event (N° of Discontinuation of Tedizolid Therapy)	N° of Episodes of Adverse Effects *
anemia (2)	4
asthenia	1
leukopenia	2
thrombocytopenia	2
headache	2
pruritus	4
abdominal pain	1
nausea/vomiting (1)	2
vertigo	1
xerosis	1
dysgeusia	1
epistaxis	1
arthralgia (1)	2
thrush	1
insomnia	2
intermittent blurred vision	1
Total	28

* Five patients had more than one episode of adverse effects.

**Table 4 antibiotics-10-00004-t004:** Adverse events according to the duration and the antibiotic regimen in 33 patients treated with tedizolid.

Patients’ Characteristics	N° of Patients (%), Total = 33	*p*
≥1 adverse event	20 (60.6)	
Any combination therapy - Yes (*n* = 18) - No (*n* = 15)	11 (61.1) 9 (60)	0.8
Rifampicin combination therapy - Yes (*n* = 16) - No (*n* = 17)	9 (56.3) 11 (64.7)	0.9
Duration of treatment ≤6 weeks - Yes (*n* = 13) - No (*n* = 20)	7 (53.8) 13 (65)	0.8

## Data Availability

Data are available upon request to Pr Eric Senneville (esenneville@ch-tourcoing.fr); the French authorities do not authorize the sharing of this type of data without prior consent and infor-mation of the patients on their final use.

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
