# Peer review of "Tolerance of Prolonged Oral Tedizolid for Prosthetic Joint Infections: Results of a Multicentre Prospective Study"

_antibiotics, 2020, doi:10.3390/antibiotics10010004_

Round 1
Reviewer 1 Report
This is an important article, as it discusses a condition which is difficult to treat, particularly in a vulnerable population like the cohort studied here. The authors are right to omit from the results the final outcome of the treatment, but it would be interesting to note how successful this treatment was.
The tables should be explained in the text, and not just referred to. Table 3 in its present form is not publishable. The authors should find a way to present a concise version, and put the full table as an appendix, available to the reader upon request.
The microbiological work-up should include sensitivity data for the study drug for all isolates (MIC and MBC). Drug blood levels would enrich this paper with valuable data.
Author Response
REVIEWER # 1 Antibiotics |
Yes |
Can be improved |
Must be improved |
Not applicable |
Does the introduction provide sufficient background and include all relevant references? |
(x) |
( ) |
( ) |
( ) |
Is the research design appropriate? |
(x) |
( ) |
( ) |
( ) |
Are the methods adequately described? |
( ) |
(x) |
( ) |
( ) |
Are the results clearly presented? |
( ) |
( ) |
(x) |
( ) |
Are the conclusions supported by the results? |
(x) |
( ) |
( ) |
( ) |
Comments and Suggestions for Authors
- This is an important article, as it discusses a condition which is difficult to treat, particularly in a vulnerable population like the cohort studied here. The authors are right to omit from the results the final outcome of the treatment, but it would be interesting to note how successful this treatment was.
Authors’ response :
We added a mention to the failure cases at one-year follow-up in the Results section
- The tables should be explained in the text, and not just referred to. Table 3 in its present form is not publishable. The authors should find a way to present a concise version, and put the full table as an appendix, available to the reader upon request.
Authors’ response :
We modified Table 3 by dividing it in two distincts tables (3 and 4)
We added sentences in the Results section to comment the data from table 3 and 4
Ex Table 3 is mentionned in the text as an appendix available on request
- The microbiological work-up should include sensitivity data for the study drug for all isolates (MIC and MBC). Drug blood levels would enrich this paper with valuable data.
Authors’ response : We added data on the geometric mean MIC values of linezolid (not avalable for tedizolid) for staphylococci, streptococci and enterococci in the Results section. TZD blood levels are not available.
|
Reviewer 2 Report
Dear Authors,
In the present study you reported findings on the tolerance of prolonged oral tedizolid for prosthetic joint infections.
The topic is very interesting, and the study is well written consistent with the Journal purposes.
The article does not require a revision of the English language by a native speaker and the Instructions for Authors have been respected (the figures are of good quality and the tables are well made).
However, Introduction Section should be improved to better focus on the management of knee and hip osteoarthritis, clarifying when there is a need of a total joint replacement.
You should start the Introduction better clarifying the definition and diagnosis of osteoarthritis (taking into account that 32 out 33 patients underwent total knee or hip replacement) and the potential therapeutic conservative approaches before surgery citing the following references:
- Martel-Pelletier J, et al. Cartilage in normal and osteoarthritis conditions. Best Pract Res Clin Rheumatol 2008; 22: 351-84. doi: 10.1016/j.berh.2008.02.001.
- Iolascon G, et al. Early osteoarthritis: How to define, diagnose, and manage. A systematic review. 2017 Nov;8(5-6):383-396. doi: 10.1016/j.eurger.2017.07.008.
- Rabini A, et al. Effects of focal muscle vibration on physical functioning in patients with knee osteoarthritis: a randomized controlled trial. Eur J Phys Rehabil Med. 2015 Oct;51(5):513-20.
- de Sire A, et al. Long-term effects of intra-articular oxygen-ozone therapy versus hyaluronic acid in older people affected by knee osteoarthritis: A randomized single-blind extension study. J Back Musculoskelet Rehabil. 2020 Feb 28. doi: 10.3233/BMR-181294.
- McAlindon TE, et al. OARSI guidelines for the non-surgical management of knee osteoarthritis. Osteoarthritis Cartilage 2014; 22: 363-88. doi: 10.1016/j.joca.2014.01.003.
- Migliore A, et al. The perspectives of intra-articular therapy in the management of osteoarthritis. Expert Opin Drug Deliv. 2020 Sep;17(9):1213-1226. doi: 10.1080/17425247.2020.1783234.
Author Response
REVIEWER # 2 Antibiotics |
Yes |
Can be improved |
Must be improved |
Non applicable |
Does the introduction provide sufficient background and include all relevant references? |
(x) |
( ) |
( ) |
( ) |
Is the research design appropriate? |
(x) |
( ) |
( ) |
( ) |
Are the methods adequately described? |
(x) |
( ) |
( ) |
( ) |
Are the results clearly presented? |
(x) |
( ) |
( ) |
( ) |
Are the conclusions supported by the results? |
(x) |
( ) |
( ) |
( ) |
Comments and Suggestions for Authors
Dear Authors,
In the present study you reported findings on the tolerance of prolonged oral tedizolid for prosthetic joint infections.
The topic is very interesting, and the study is well written consistent with the Journal purposes.
The article does not require a revision of the English language by a native speaker and the Instructions for Authors have been respected (the figures are of good quality and the tables are well made).
Authors’ response :
We thank Reviewer #2 for his helpful comments
We added the following sentences in the Introduction section and some of the references suggested by Reviewer #2
Prosthetic joint infection (PJI) is a serious and complex complication following arthroplasty at an incidence rate after hip or knee replacement of 1 to 2% [1]. The aims of the management of patients with PJIs are to restore a satisfactory joint function and to eliminate the infection. The surgical options include debridement antibiotics and implant retention (DAIR), one or two-stage replacement, arthroplastic resection and sometimes amputation. Given the increasing burden of these infections especially on the elderly population, developing new therapies such as cell therapy for preventing the progression of osteo-arthritis and thus the need for total joint arthroplasty is an important field of research [2-5].
- Li C, Renz N, Trampuz A. Management of Periprosthetic Joint Infection. Hip Pelvis. 2018 Sep;30(3):138-146
- de Sire A, et al. Long-term effects of intra-articular oxygen-ozone therapy versus hyaluronic acid in older people affected by knee osteoarthritis: A randomized single-blind extension study. J Back Musculoskelet Rehabil. 2020 2020;33(3):347-354.
- McAlindon TE, et al. OARSI guidelines for the non-surgical management of knee osteoarthritis. Osteoarthritis Cartilage 2014; 22: 363-88.
- Migliore A, et al. The perspectives of intra-articular therapy in the management of osteoarthritis. Expert Opin Drug Deliv. 2020 Sep;17(9):1213-1226.
Round 2
Reviewer 1 Report
After the authors have made the changes, the paper is ready for publication.